# Antioxidative and Mitochondrial Protection in Retinal Pigment Epithelium: New Light Source in Action

**DOI:** 10.3390/ijms24054794

**Published:** 2023-03-01

**Authors:** Ming Jin, Xiao-Yu Zhang, Qian Ying, Hai-Jian Hu, Xin-Ting Feng, Zhen Peng, Yu-Lian Pang, Feng Yan, Xu Zhang

**Affiliations:** Jiangxi Provincial Key Laboratory for Ophthalmology, Jiangxi Clinical Research Center of Ophthalmic Disease, Affiliated Eye Hospital of Nanchang University, 463 Bayi Road, Nanchang 330006, China

**Keywords:** low color temperature, phosphor-free light-emitting diodes (LEDs), white light-emitting diodes (LEDs), phototherapy, age-related macular degeneration (AMD), retinal pigment epithelium (RPE), mitochondria, reactive oxygen species (ROS)

## Abstract

Low-color-temperature light-emitting diodes (LEDs) (called 1900 K LEDs for short) have the potential to become a healthy light source due to their blue-free property. Our previous research demonstrated that these LEDs posed no harm to retinal cells and even protected the ocular surface. Treatment targeting the retinal pigment epithelium (RPE) is a promising direction for age-related macular degeneration (AMD). Nevertheless, no study has evaluated the protective effects of these LEDs on RPE. Therefore, we used the ARPE-19 cell line and zebrafish to explore the protective effects of 1900 K LEDs. Our results showed that the 1900 K LEDs could increase the cell vitality of ARPE-19 cells at different irradiances, with the most pronounced effect at 10 W/m^2^. Moreover, the protective effect increased with time. Pretreatment with 1900 K LEDs could protect the RPE from death after hydrogen peroxide (H_2_O_2_) damage by reducing reactive oxygen species (ROS) generation and mitochondrial damage caused by H_2_O_2_. In addition, we preliminarily demonstrated that irradiation with 1900 K LEDs in zebrafish did not cause retinal damage. To sum up, we provide evidence for the protective effects of 1900 K LEDs on the RPE, laying the foundation for future light therapy using these LEDs.

## 1. Introduction

Today, light-emitting diodes (LEDs) are widely used because of their low power consumption, ease of use, long service life, and good color rendering index (CRI) scores. However, the harmful blue band portion of phosphor white LEDs may affect human health in two directions: (1) potential retinal phototoxicity [1,2]; (2) disruption of internal circadian rhythm, including health problems associated with circadian rhythm disturbances [3,4,5,6,7,8,9]. 

To avoid blue-light hazards, LED developers abandoned the principle of using blue LEDs to excite phosphors for generating white light; they instead use silicon substrate InGaN yellow light and AlGaInP red LEDs to synthesize blue-free low-color-temperature phosphor-free LEDs (called 1900 K LEDs for short). The CRI of 1900 K LEDs can exceed 80, and the color temperature is 1600–2200 K, which meets lighting needs.

Shortly after the emergence of 1900 K LEDs, we conducted relevant retinal phototoxicity in vitro experiments using these LEDs. We found that the 1900 K LEDs caused less cell death than other high-color-temperature phosphor white LEDs [10]. We also found that the 1900 K LEDs benefited wound healing, hair growth, and melatonin and glutamate secretion. Moreover, the 1900 K LEDs could also protect the ocular surface [11]. The above experiments proved that 1900 K LEDs may have a certain protective effect and high safety.

By 2040, the number of people with age-related macular degeneration (AMD) will be nearly 300 million [12], posing a major public health problem with a significant socioeconomic impact. It is well known that progressive degeneration and death of the retinal pigment epithelium (RPE) represent key pathological processes in AMD. Therefore, treatment for RPE is significant [13,14]. Considering the protective effects of 1900 K LEDs on the ocular surface, we hypothesized that the 1900 K LEDs may protect the RPE. Actually, excess reactive oxygen species (ROS) and mitochondrial defects exist in the RPE of AMD [15]. Red–yellow light-related photobiomodulation (PBM) treatment precisely targets the mitochondria and oxidative stress [16,17,18,19]. Therefore, we believe that 1900 K LEDs can play a key role in preventing and treating AMD by protecting the RPE. ARPE-19 cells, a reliable and standard model for AMD research, were mainly used as the experimental object in this study to preliminarily investigate the protective effects of 1900 K LEDs on RPE. In addition, we preliminarily demonstrate in vivo that the irradiation of this new light source posed no harm to the retina in zebrafish. Specific experiments included a cell activity assay, cell death assay, ROS level detection, mitochondrial imaging, mitochondrial DNA (mtDNA) damage detection, Western blotting, and hematoxylin and eosin (H&E) staining.

## 2. Results and Discussion

### 2.1. 1900 K LEDs Regulate Cell Activity According to Irradiance and Light Time

After irradiation at 10 W/m^2^ for 48 h, all LEDs except the 1900 K LEDs decreased cell activity (Figure 1a). The 1900 K LEDs at three irradiances (5, 10, and 15 W/m^2^) all increased cell activity to varying degrees, with the irradiance at 10 W/m^2^ increasing cell activity most significantly (Figure 1b). Thus, the irradiance was kept at 10 W/m^2^. After 1, 2, and 3 days of irradiation with 1900 K LEDs, the cell activity increased by about 30%, 70%, and 75%, respectively (Figure 1c). These experiments led us to determine the parameters of LED illumination in later experiments, i.e., 10 W/m^2^ illumination for 48 h. Such experiments on the effects of LEDs with different color temperatures on cells are not uncommon. Chen et al. [20] found that 7378 K LEDs caused excessive intracellular ROS production and severe DNA damage in cultured human lens epithelial cells (hLECs) compared to 2954 K and 5624 K LEDs, which led to G2/M phase block and apoptosis. Another study found that 7378 K LEDs, but not 2954 K LEDs, induced the upregulation of VEGF-A, IL-6, and IL-8, as well as the downregulation of MCP-1, through an accumulation of ROS and the activation of MAPK and NF-κB signaling pathways [21]. These studies partially support our results. However, the subjects of these experiments did not involve LEDs with such a color temperature of 1900 K. To address this gap, we designed follow-up experiments.

### 2.2. Pretreatment with 10 W/m^2^ 1900 K LEDs Reduced the Death of ARPE-19 Cells

To explore the protective effects of the 1900 K LEDs on damaged cells, we chose the hydrogen peroxide (H_2_O_2_) damage model, which is classically used for the study of AMD. As shown in Figure 2a, we treated cells with a series of concentrations of H_2_O_2_ (200 μM, 400 μM, 600 μM, 800 μM, and 1000 μM) for 6 h before choosing 400 μM as the final concentration. Next, we investigated the effects of two paradigms on ARPE-19 cells and found that the protection of the light post-treatment was limited (Figure 2b). This was also corroborated by the bright-field images and the flow cytometry results on apoptosis (see Appendix A). Therefore, we switched to a 1900 K LED pre-illumination paradigm for 48 h, followed by 400 μM H_2_O_2_ damage for 6 h. To verify that the protective effect of these LEDs was not transient, we added several detection points (3 h, 9 h, and 24 h) after the removal of H_2_O_2_. The cell activity of the 1900 K + H_2_O_2_ group was higher than that of the H_2_O_2_ group at four time points (Figure 2c–f). Interestingly, we observed no significant difference in cell morphology and apoptosis among the four groups at 0 h after the removal of H_2_O_2_ (see Appendix A). We posit that 400 μM H_2_O_2_ damage for 6 h did not cause cell apoptosis but led to a decrease in cell activity. The 1900 K LEDs prevented the decrease in cell activity caused by H_2_O_2_. The negative manifestation of cells after a blow tends to emerge later. To investigate the protection range of LEDs at 10 W/m^2^ irradiance, we detected the cell state and apoptosis at 1000 μM H_2_O_2_ concentration in the light pretreatment paradigm. The number of cells in the 1900 K + H_2_O_2_ group was more than that in the H_2_O_2_ group. Furthermore, the connection between the cells in the H_2_O_2_ group almost disappeared. However, there were still some cell connections among the cells in the 1900 K + H_2_O_2_ group, and the percentage of cells with an increased refractive index around the cells was also lower than that in the H_2_O_2_ group. The flow cytometry results also confirmed this; the apoptosis rate of the 1900 K + H_2_O_2_ group was significantly lower than that of the H_2_O_2_ group (Figure 2g–i). In fact, we extended the time of H_2_O_2_ injury to 24 h, and 10 W/m^2^ LED illumination for 48 h also produced a protective effect on the cells (see Appendix A). In conclusion, preliminary experiments suggest that the 1900 K LEDs likely had cytoprotective effects.

Before us, there were also many experiments in which red lasers, LEDs, or even compound LEDs were used to protect the retinas of animals or volunteers [17,22,23,24,25,26,27]. Some studies suggested that the sequence of light treatments would affect treatment outcomes. Albarracin et al. [25] found that 670 nm LED light treatment before and during exposure to damaging white light significantly ameliorated the glare-induced reduction in photoreceptor function. However, photoreceptor function in animals treated with light following intense light-induced injury initially decreased but recovered 1 month after exposure. The above results suggest that phototherapy pretreatment and period treatments may have earlier or greater protective effects, partially consistent with our results. Regarding the different phenomenon in the 1900 K + H_2_O_2_ and H_2_O_2_ + 1900 K paradigms, our explanation is as follows: compared with other color temperature LEDs, the 1900 K LEDs were able to increase the cell activity of ARPE-19 cells. The interpretation of cell activity results usually requires two aspects; one is cell proliferation, and the other is the alteration of intracellular mitochondrial dehydrogenase activity. We cannot deny that 1900 K LEDs could promote ARPE-19 cell proliferation to a certain extent (see Appendix A). We also proved that 1900 K LEDs could increase intracellular activities. The 4000 K and blue-light LEDs elevated intracellular ROS levels, while the 1900 K LEDs reduced them. In addition, mitochondrial imaging and DNA damage experiments showed that 1900 K LEDs could reduce the damage to mitochondria caused by H_2_O_2_. Coupled with the above results and the intracellular protein expressions of nuclear factor E2-related factor 2 (NRF2), heme oxygenase-1 (HO-1), microtubule-associated protein 1 light chain 3 beta (LC-3B), dynamin-related protein 1 (DRP1), and optic atrophy protein 1 (OPA1), we assume that pre-irradiation with 1900 K LEDs not only promoted ARPE-19 cell proliferation but also reduced the damage caused by H_2_O_2_, which led to the obvious increase in cell activity in the 1900 K group. However, if the cells are first damaged with H_2_O_2_, many protective effects may not work under severe trauma [28].

The 1900 K LEDs can also be used for illumination, while most light sources used for treatment in other studies cannot be used for illumination. We assume that 1900 K LEDs play a role in preventing AMD while simultaneously providing illumination, which can greatly improve patient compliance and, most importantly, reduce the cost of treatment.

### 2.3. The 1900 K LEDs Play an Active Role in Antioxidative Stress of ARPE-19 Cells

Considering the preliminary proof that 1900 K LED pretreatment could protect ARPE-19 cells, we wanted to explore the possible mechanism. First, we focused on antioxidative stress. We found that red intranuclear fluorescence representing intracellular ROS level increased as the color temperature of LEDs increased (Figure 3a). However, the ROS level of the 1900 K group was much lower than that of the control (Figure 3b,c). We then repeated this experiment in the light pretreatment paradigm. As expected, the ROS level of the 1900 K group was still lower than that of the control, while the situation in the H_2_O_2_ group was the opposite. The ROS level of the 1900 K + H_2_O_2_ group was between that of the H_2_O_2_ group and the 1900 K group (Figure 3d,e). This result is not surprising. Shen et al. [17] also found that 670 nm LEDs could reduce the level of ROS in the mitochondria and enhance mitochondrial function in rat primary Müller cells. In the light pretreatment paradigm, we found that NRF2 in the H_2_O_2_ group was upregulated, and HO-1 also showed an upregulation trend. NRF2 in the 1900 K group also increased slightly. Interestingly, HO-1 protein decreased in the 1900 K group. The expression of HO-1 in the 1900 K + H_2_O_2_ group was lower than that in the H_2_O_2_ group. However, the expression of NRF2 was similar to that in the control group (Figure 3f,g). NRF2 is the upstream protein of HO-1. As an inducible enzyme, HO-1 is a measurable indicator of oxidative stress and is upregulated with NRF2 protein [29]. Núñez-Álvarez et al. [30] also demonstrated that red light (625–635 nm) at 6.5 W/m^2^ could reduce the HO-1 caused by blue light. A recent study showed that direct knockdown or pharmacological inhibition of HO-1 significantly blocked oxidative stress-induced ferroptosis in RPE cells [31]. Electrophile reagents and ROS disrupt the interaction between CncC and Keap1 after stress exposure. CncC is not degraded but accumulates in the nucleus, forms heterodimers with Maf-S, binds to AREs, and activates target gene transcription [32,33]. Therefore, after H_2_O_2_ injury, NRF2 enters the nucleus and binds to AREs to activate the transcription of target genes, resulting in the upregulation of HO-1. In contrast, 1900 K LEDs can upregulate the expression of NRF2. However, the relatively low intracellular ROS level does not allow NRF2 and Keap1 to disengage into the nucleus to activate downstream genes. It is only during stress that upregulated NRF2 enters the nucleus and increases the cell’s ability to resist oxidative stress. Our ROS results also prove this point. Our experiments preliminarily show that the 1900 K LEDs play a protective role by increasing the nonenzymatic antioxidant system, not the enzymatic one. Considering the effect of 1900 K LEDs on the mitochondria, we suspected the most likely candidate was a coenzyme or glutathione. 

In addition, oxidative stress can cause an increase in stress-induced autophagy in cells. Therefore, we detected the expression of autophagy-related protein LC3B and found that 400 μM H_2_O_2_ could increase the ratio of LC3B-II/LC3B-I, thereby increasing autophagy in ARPE-19 cells. The ratio of LC3B-II/LC3B-I in the 1900 K group was not different from that in the dark group. The autophagy level in the 1900 K+ H_2_O_2_ group was much lower than that in the H_2_O_2_ group (Figure 3h). A previous study used the same H_2_O_2_ concentration of 400 μM and found that the ratio of LC3B-II/LC3B-I in RPE cells was upregulated after H_2_O_2_ exposure for 6 h compared with 0 h [34]. The study of Hu et al. also supports our results [35]. Altogether, this suggests that 1900 K LEDs can reduce the level of ROS to protect RPE from death.

### 2.4. The 1900 K LEDs Can Reduce Mitochondrial Damage Caused by H_2_O_2_ in ARPE-19 Cells

Mitochondrial dysfunction is implicated in the pathophysiology of several age-related diseases, including AMD. Mitochondrial fission, fusion, and mitophagy are important components of mitochondrial quality control [36]. Therefore, the modulation of mitochondrial dynamics may be a valuable strategy for treating retinal degenerative diseases such as AMD. We previously proved that 1900 K LEDs can reduce the level of ROS in ARPE-19 cells and that mitochondria represent the main target of ROS. Therefore, we shifted the spotlight toward the effects of the 1900 K LEDs on mitochondrial function. First, we performed mitochondrial imaging, adopting the experimental parameters from the study of Jang et al. [37]. They used 0.5 mM H_2_O_2_ to act on ARPE-19 cells for 1 h and observed that mitochondrial fission increased, whereas PARP-1 inhibitors could reduce the degree of fission in ARPE-19 cells. Fluorescence images showed that the mitochondria in the dark group and 1900 K group were linear and reticular in shape. Mitochondria in the shape of short rods, dots, or spheres could be observed after 400 μM H_2_O_2_ injury, which could be rescued by 1900 K LEDs (Figure 4a). Then, we quantified the morphology of the mitochondria. We found that H_2_O_2_ at the present experimental concentration caused the long, branching mitochondria to be broken into different parts, resulting in an increase in the number of mitochondrial networks but a decrease in the network size (number of branches). This led to an increase in the number of individual mitochondria, as well as a reduction in branch length. A similar phenomenon was observed in [38,39,40]. The 1900 K LEDs could mitigate this phenomenon. However, it is interesting that the 1900 K LEDs also increased mitochondrial fragmentation (Figure 4b–e). Combined with the previous results on autophagy, we do not believe that the 1900 K LEDs caused a negative increase in mitochondrial fragmentation because the 1900 K LEDs did not cause an increase in cellular autophagy but rather accelerated the fusion cleavage cycle due to the increase in mitochondrial exchange.

In vitro studies have shown that H_2_O_2_-induced oxidative stress leads to RPE cell death by causing preferential damage to their mitochondrial DNA [41,42]. Moreover, mitochondrial DNA damage is more extensive and longer-lasting than nuclear DNA damage in human cells after oxidative stress [43]. Therefore, we can determine the degree of oxidative stress by detecting mitochondrial DNA damage. Our results show that mitochondrial DNA in ARPE-19 cells was damaged after 400 μM H_2_O_2_ injury for 6 h. Irradiation with 1900 K LEDs did not cause mitochondrial DNA damage in cells but reduced the mitochondrial DNA damage caused by H_2_O_2_ (Figure 4f,g). Combined with the results on ROS levels, it is not difficult to conclude that 1900 K LEDs protect the mitochondrial DNA from damage by reducing intracellular ROS. Indeed, the mitochondrial genome is susceptible to oxidative stress damage, and defects in the mitochondrial DNA repair pathway are important factors in the pathogenesis of retinal degeneration. The use of therapeutic modalities that specifically target the mitochondria to protect them from oxidative stress or promote mtDNA repair may provide a potential alternative for treating retinal degenerative diseases such as AMD [44]. PBM has been shown to enhance mitochondrial activity and restore the function of damaged mitochondria, thus promoting cell survival in vitro by stimulating the activity of cytochrome c oxidase [45]. Gopalakrishnan et al. [46] showed that far-infrared PBM improved functional and structural outcomes in animal models of retinal damage and degenerative diseases. A short near-infrared PBM process would preserve mitochondrial metabolic status and reduce photoreceptor loss. Recently, 670 nm LEDs have been used in clinical trials for treating macular edema in humans. The 670 nm LEDs could not only enhance the mitochondrial membrane potential of rodent photoreceptors but also protect Müller cells and photoreceptors from damage in vivo. Moreover, in vitro experiments demonstrated that 670 nm LEDs could enhance mitochondrial function and protect cultured Müller cells from oxidative stress [17].

According to mitochondrial morphology experiments, we detected the mitochondrial fission and fusion proteins. We found that the DRP1 protein was downregulated in the H_2_O_2_ group but upregulated in the 1900 K groups. The DPR1 level of the 1900 K + H_2_O_2_ group was between that of the H_2_O_2_ group and the 1900 K group (Figure 4h). For OPA1, an endomembrane fusion protein, the total protein levels of the four groups were not significantly different (Figure 4i). However, the ratio of L-OPA1/S-OPA1 in the H_2_O_2_ group showed a downward trend. The opposite was observed in the 1900 K group. Similar to DRP1, the ratio of L-OPA1/S-OPA1 in the 1900 K + H2O2 group was also between that in the H_2_O_2_ and 1900 K groups (Figure 4j). Our previous hypothesis was that H_2_O_2_ would upregulate intracellular DRP1 levels, leading to mitochondrial fragmentation. However, the level of DRP1 did not increase after H_2_O_2_ treatment; thus, it cannot be excluded that DRP1 may still play a role in oxidation-induced mitochondrial fragmentation [40]. Interestingly, there was an increase in DRP1 in the 1900 K group. Sheng et al. also observed the upregulation of OPA1 and FIS1 in the zebrafish retina after resveratrol treatment [47]. FIS1 is also a mitochondrial fission protein. Under physiological conditions, mitochondrial fusion and fission occur in a dynamic balance, which is altered by external stimulation. The mitochondrial morphology in the 1900 K group was linear and reticular, as also seen in the control group, and the mitochondrial DNA was not damaged; thus, DRP1 levels in the 1900 K group were likely upregulated to balance mitochondrial fusion. The increase in the fusion split cycle would likely promote communication between mitochondria and increase mitochondrial function [48], consistent with the results of the previous mitochondrial morphology experiments.

Several studies support our results regarding the OPA1 protein. Li et al. irradiated R28 cells with blue light for different times and found that the total protein level of OPA1 did not change at different time points (0–24 h) [49]. Garcia et al. also found that the overall expression of OPA1 was not altered by H_2_O_2_ treatment, but the L-OPA1 isoform in both H9c2 and 143B cell lines was degraded [40]. When the transmembrane potential of the inner mitochondrial membrane is intact, the L-OPA1 isoform is necessary for the process of mitochondrial inner membrane fusion. When the transmembrane potential is lost, L-OPA1 is cleaved by the stress-sensitive overlapping with the m-AAA protease 1 homolog (OMA1) metalloprotease into a short S-OPA1 isoform that cannot promote inner mitochondrial membrane fusion and may even participate in inner mitochondrial membrane fission, resulting in the collapse of the mitochondrial network into fragmented organelle populations [50,51,52]. This is an organismal protection mechanism. If a cell encounters stress above a critical threshold, severely damaged mitochondria can contaminate other mitochondria by rejoining the mitochondrial network before being eliminated by autophagy. The organism has to prevent this from happening. At this time, OMA1 is rapidly activated by a low membrane potential and low levels of adenosine triphosphate (ATP) to cleave L-OPA1. Even in the absence of L-OPA1 or with very low membrane potential, the outer membranes of these mitochondria can still fuse. However, the inner membranes do not, resulting in several inner membrane matrix compartments surrounded by a common outer membrane, akin to peas in a pod, waiting to be eliminated by autophagy [53]. Wang et al. observed a similar phenomenon [39]. Accordingly, we believe that 1900 K LEDs may protect the mitochondria by reducing the stress caused by H_2_O_2_, thereby reducing OMA1 activation and L-OPA1 cleavage. The above explanation requires further research to confirm the regulatory effects of protective factors on OPA1 and DRP1.

### 2.5. Irradiation with 1900 K LEDs Did Not Cause Retinal Damage in Zebrafish

Zebrafish show great similarities with humans in terms of eye shape, anatomy, gene expression, and function, and they have the advantages of rapid development, in vitro fertilization, transparent embryos, and easy observation. Therefore, they play an important role in research on eye development and diseases [54,55]. In addition, zebrafish have no eyelids; hence, experiment failure caused by eyelids blocking the light can be avoided. We analyzed the changes in the zebrafish retina after exposure to the 1900 K and blue-light LEDs. Compared with the control group, the thickness of the outer nuclear layer (ONL), photoreceptor layer (PRL), and RPE was significantly decreased in the blue light group on day 7, whereas the thickness of ONL, PRL, and RPE in the 1900 K LED light group was not significantly altered (Figure 5). This in vivo experiment preliminarily proves the safety of 1900 K LEDs in the fundus of zebrafish.

In recent years, PBM has been recognized as an emerging therapy for treating age-related degenerative and mitochondrial-related diseases [56,57,58]. The main mechanism may be the promotion of cytochrome c oxidase (COX) activity and ATP content. PBM can also promote the release of nitric oxide (NO) from intracellular storage areas (e.g., heme-containing proteins) to enhance its bioavailability, triggering the activation of subsequent protective signaling pathways [57]. Although the light used in PBM previously had to be coherent and polarized, e.g., the light produced by He–Ne lasers, these properties are no longer considered essential. LED arrays can also be used as light sources in photomedicine [59], whereby monochromatic and multi-wavelength LEDs affect different cellular targets with potentially additional benefits [26,27]. The advantages and disadvantages of PBM with laser or LEDs as the radiation source remain controversial. Lasers are coherent, the scattering, absorption, and filtering due to the eye structure are reduced, and the area and dose of treatment can be controlled more precisely. LEDs are safer, less costly, and more readily available, making them a new therapeutic light source with more potential than lasers. However, establishing treatment parameters is more problematic in PBMs using LEDs as radiation sources. There are still many problems with LED-mediated PBMs. Our results further broaden the range of LEDs that can produce cytoprotective effects, and they demonstrate that light sources of PBM are no longer limited to monochromatic LEDs but can also include composite LEDs with illumination functions.

In addition, most research on PBM involved basic phenotypic studies. A recent study found that a 632.8 nm red laser reduced the amyloid β protein (Aβ) deposits in Alzheimer’s disease models by activating the PKA/Sirtuin 1 (SIRT1) signaling pathway [60]. This gives us great inspiration; many studies have proven the effectiveness of PBM. Nevertheless, research on the physiological mechanism has been limited; therefore, it will be necessary to explore the signaling pathways in PBM in the future. Accordingly, we examined the effect of the 1900 K LEDs on SIRT1 expression, finding that H_2_O_2_ could upregulate the expression of SIRT1 and that the 1900 K LEDs increased the amount of SIRT1 more than the H_2_O_2_ group (see Appendix A). We also found the increased expression of other members of the SIRT family, such as Sirtuin 3 and Sirtuin 5, after the 1900 K LED treatment (Xu Zhang, unpublished data). We hypothesize that the LEDs might bring some positive effects to the cells by upregulating the sirtuin family. This mechanism is worthy of in-depth studies in the future. As a limitation of this study, we only preliminarily discussed the protective effects of the 1900 K LEDs on RPE cells and zebrafish; further in vivo experiments and related specific mechanisms are lacking.

Our study demonstrated that pretreatment with the 1900 K LEDs could counteract H_2_O_2_ damage to RPE cells by reducing the generation of oxidative stress and mitochondrial damage in ARPE-19 cells (Figure 6). The 1900 K LEDs may need to be in a certain range of irradiance for optimal protection, which is likely cumulative. In addition, irradiation with the 1900 K LEDs for 7 days did not cause retinal damage in zebrafish. Our research is very meaningful because protecting cells from death is a definite and effective measure in the absence of allopathic treatment. This is the first study to investigate the protective effects of 1900 K low-color-temperature phosphor-free LEDs on RPE and zebrafish. The noninvasiveness, good compliance, convenience, and availability of light therapy using the 1900 K LEDs would make it the most promising emerging tool for preventing and treating AMD. Moreover, light therapy can also provide inspiration for the management of other similar diseases, which is of great benefit to human beings.

## 3. Materials and Methods

### 3.1. Cell Culture

We purchased the ARPE-19 cells from the American Type Culture Collection (ATCC, Manassas, Virginia). The cells were cultured in Dulbecco’s modified Eagle medium (DMEM/F-12, BI, Israel) supplemented with 10% (*v*/*v*) fetal bovine serum (FBS, BI, Israel), 100 U/mL penicillin, and 100 μg/mL streptomycin (Solarbio, Beijing, China) in a cell incubator at 37 °C with 5% CO_2_. Cell passages were performed every 4 to 5 days. We used ARPE-19 cells from passages 3–10 in our experiments.

### 3.2. Animals

All wild-type zebrafish (*Danio rerio*) of the AB strain were obtained from the China Zebrafish Resource Center, CZRC (Wuhan, China). Zebrafish were maintained in 14 h light/10 h dark cycles at 28.5 °C. The use and manipulation of zebrafish were approved by the ethical review committee, and the study adhered to the ARVO Statement for the Use of Animals in Ophthalmic and Vision Research.

### 3.3. Light Treatment and H_2_O_2_ Damage

This study mainly used the following LEDs: 1900 K, 4000 K, 6600 K, and blue-light LEDs. The light source and lighting device were the same as described in our previous article (Figure 7a,b) [10]. The entire light box was placed in a cell incubator to ensure normal cell growth. The top panel in Figure 7c shows the light post-treatment paradigm, which involves damaging the cells with 400 μM H_2_O_2_ for 6 h and then illuminating the cells using the 1900 K LEDs at 10 W/m^2^ irradiance for 48 h. The lower panel of Figure 7c shows the light pretreatment paradigm. The light pretreatment paradigm involved exposing the cells to the 1900 K LEDs at 10 W/m^2^ for 48 h and then damaging them using 400 μM H_2_O_2_ for 6 h. 

### 3.4. Light Exposure in Animal Experiments

Zebrafish were maintained in a transparent aquarium (10 cm × 10 cm × 10 cm), and each aquarium was placed in the center of a light box with reflective interior walls (12 cm × 12 cm × 12 cm). The LEDs were placed on the four side walls of the light box to improve the directional uniformity of the radiation. The light exposure was consistent with the circadian rhythm of zebrafish automatically, and the light intensity was 3000 lux.

### 3.5. Cell Activity Assay

We measured the cell activity using the Cell Counting Kit-8 (CCK-8) (40203ES60, Yeasen, Shanghai, China) according to the reagent instructions. ARPE-19 cells were seeded into 96-well plates at a density of 3 × 10^3^ cells per well and then incubated for 24 h. Subsequently, cells received the corresponding treatment. Next, the cells were cultured in a medium supplemented with 10% (*v*/*v*) WST-8 for 2 h. The absorbance was measured by a microplate reader (Multiskan Mk3, Thermo Scientific, Shanghai, China).

### 3.6. Cell Death Assay

After treatment, the cells were immediately observed under a microscope, and three random fields were recorded. Afterward, we detected apoptosis using an Annexin V–FITC/PI Apoptosis Detection Kit (40302ES50, Yeasen, Shanghai, China). The cells were digested with EDTA-free trypsin, collected, and centrifuged at 300× *g* at 4 °C for 5 min. After washing twice with PBS, the cells were resuspended in the buffer. Then, a staining solution was added to the cells before incubating them for 10–15 min in the dark at room temperature. After incubation, 1× Binding Buffer was added, and the results were detected using a flow cytometer (DxFLEX, Beckman, Suzhou, China).

### 3.7. ROS Levels Detection

According to the instructions, we incubated cells with the DHE medium mixture diluted to 100 μM in the dark at 37 °C for 20 min. Then, we washed the cells with DHE staining solution and randomly photographed three fields of view. In addition, we used the same reagent to quantitatively analyze the level of intracellular ROS. After being stained with DHE, the cells were collected and washed twice, resuspended with PBS, and detected by flow cytometry (DxFLEX, Beckman, Suzhou, China).

### 3.8. Mitochondrial Imaging

ARPE-19 cells were seeded into 35 mm confocal dishes at a density of 4 × 10^5^ cells and incubated for 24 h until their confluence reached 50–70%. Then, we transfected ARPE-19 cells using DsRed2-Mito plasmid (Wuhan Miaoling Biotechnology Co., Ltd. Wuhan, China) to label the mitochondria according to the instructions of the Hanheng LipoFiter3 Liposome Transfection Reagent (HB-TRLF3-1000, Hanheng, Shanghai, China). The DsRed2-Mito plasmid encodes a fusion of Discosoma sp. red fluorescent protein (DsRed2; 1, 2) and the mitochondrial targeting sequence from subunit VIII of human cytochrome c oxidase (Mito; 3, 4). Next, 36 h after transfection, cells were irradiated with the 1900 K LEDs for 48 h before adding 400 μM H_2_O_2_ for 3 h. The morphology of the mitochondria was observed under a Zeiss confocal microscope (LSM800, ZEISS, Göttingen, Germany), and multiple fields were randomly selected for photographing and recording. Then, we assessed the number and morphology of mitochondria using the Mitochondrial Network Analysis (MiNA) toolset.

### 3.9. Western Blotting

According to the manufacturer’s instructions for the radioimmune precipitation assay buffer (RIPA) (Solarbio, Beijing, China) and bicinchoninic acid (BCA) protein assay kit (Solarbio, Beijing, China), we lysed ARPE-19 cells and detected the protein concentration. An equal amount of protein (25 μg) was separated by electrophoresis on 10% or 12% SDS polyacrylamide gels and transferred to polyvinylidene fluoride membranes (PVDF, Millipore, United Kingdom). Then, the membranes were blocked with 5% nonfat milk for 1 h and incubated with the appropriate primary antibody (Table 1 and Appendix A) overnight at 4 °C. On the next day, the membranes were washed and incubated with the secondary antibody at room temperature for 1 h. After washing the membrane, the target proteins were detected using the EasySee Western Blot Kit (TRANS, Beijing, China), and digital images were obtained. The bands were quantified by integrating the pixel intensity using ImageJ software. β-tubulin served as an internal control.

### 3.10. Detection of mtDNA Damage

We referred to the method of Sheng et al. to detect mitochondrial DNA damage [47]. After the cells were processed, we extracted the total cell DNA and amplified the long and short mitochondrial fragments from 15 ng of total DNA following the steps of the Ezup Column Animal Genomic DNA Extraction Kit (Sangon Biotech, Shanghai, China) and the Long Amplification Taq polymerase kit (P101-d1, Vazyme, Nanjing, China). The primers of the long mitochondrial fragment (16568 bp) were F: ATGATGTCTGTGTGGAAAGTGGCTGTGC and R: GGGAGAAGCCCCGGCAGGTTTGAAGC. The primers of the short mitochondrial fragment (158 bp) were F: GATTTGGGTACCACCCAAGTATTG and R: AATATTCATGGTGGCTGGCAGTA. The PCR amplification conditions for the long mitochondrial fragments were as follows: 95 °C for 5 min, followed by 17 cycles (95 °C for 30 s; 65 °C for 15 min), with a final extension step at 72 °C for 10 min. The PCR amplification conditions for the short mitochondrial fragments were as follows: 95 °C for 5 min, followed by 17 cycles (95 °C for 30 s; 60 °C for 30 s; 72 °C for 30 s), with a final extension step at 72 °C for 10 min. All long- and short-chain PCR reactions were stopped at the linear phase. The PCR products were then separated by 0.8% and 1% agarose gel electrophoresis. Lastly, the gels were visualized under UV irradiation in Image Lab 5.2.1 (SYSTEM GelDoc XR+ IMAGELAB, Bio-Rad, Hercules, CA, USA), and the bands were quantified by ImageJ.

### 3.11. Hematoxylin and Eosin Staining and Histologic Evaluation

The specific steps were the same as described in our previous article [61]. Briefly, eyeballs from zebrafish were dehydrated in a stepwise manner after internal and external fixation. Then, eyeballs were processed with xylene and embedded in paraffin. Next, the embedded eyes were cut into 4 μm thick sections before performing H&E staining. Image-pro Plus 6.0 was used to determine the thickness of ONL, PRL, and RPE in a region beginning 250 μm from the center of the optic nerve head.

### 3.12. Statistical Analysis

All results are presented as mean ± standard deviation ($\bar{x}$ ± SD). Graph Pad Prism 7.0 software was used for statistical analysis. Student’s *t*-test was used to compare the means of two groups. One-way ANOVA and the post hoc Tukey test were used for three or more groups of samples. A *p*-value < 0.05 was considered statistically significant.

## Figures and Tables

**Figure 1 ijms-24-04794-f001:**
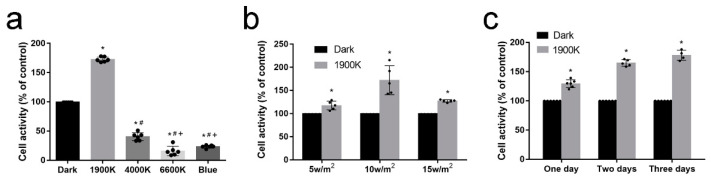
The low-color-temperature light-emitting diodes (LEDs) regulate cell activity according to irradiance and light time. (**a**) Cell activity after 48 h illumination of LEDs with different color temperatures at 10 W/m^2^; *n* = 6. * Significantly different from dark group (one-way ANOVA, *p*  <  0.0001); ^#^ significantly different from 1900 K group (one-way ANOVA, *p*  <  0.0001); ^+^ significantly different from 4000 K group (one-way ANOVA, *p*  <  0.0001). (**b**) Cell activity after 48 h illumination of the 1900 K LEDs with different radioactivity; *n* = 5. * Significantly different from dark group (two-sided, independent *t*-test: 5 W/m^2^, *p*  =  0.0032; 10 W/m^2^, *p*  =  0.0009; 15 W/m^2^, *p*  <  0.0001). (**c**) Cell activity of ARPE-19 cells after 1900 K LED illumination at 10 W/m^2^ for different times; * Significantly different from dark group (two-sided, independent *t*-test: 1 day, *n* = 6, *p*  <  0.0001; 2 days, *n* = 5, *p*  <  0.0001; 3 days, *n* = 4, *p*  <  0.0001). The data are presented as the mean ± SD.

**Figure 2 ijms-24-04794-f002:**
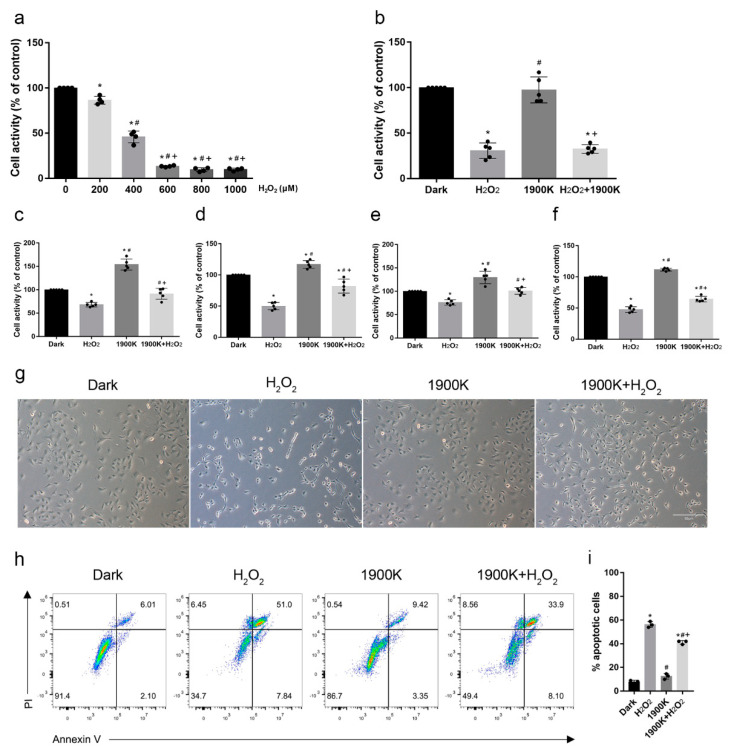
Pretreatment with the 1900 K LEDs at 10 W/m^2^ reduced the death of ARPE-19 cells. (**a**) Hydrogen peroxide (H_2_O_2_) concentration selection; *n* = 4. * Significantly different from 0 μM (one-way ANOVA: 200 μM, *p* = 0.0003; 400, 600, 800, and 1000 μM, *p*  <  0.0001); ^#^ significantly different from 200 μM (one-way ANOVA, *p*  <  0.0001); ^+^ significantly different from 400 μM (one-way ANOVA, *p*  <  0.0001). (**b**) Cell activity of ARPE-19 cells in the light post-treatment paradigm; *n* = 5. * Significantly different from dark group (one-way ANOVA: H_2_O_2_ and H_2_O_2_ + 1900 K, *p*  <  0.0001); ^#^ significantly different from H_2_O_2_ group (one-way ANOVA, *p*  <  0.0001); ^+^ significantly different from 1900 K group (one-way ANOVA, *p*  <  0.0001). Cell activity of ARPE-19 cells after removing H_2_O_2_ at 0 h (**c**), 3 h (**d**), 9 h (**e**), and 24 h (**f**) in the pre-light treatment; *n* = 5. * Significantly different from dark group (one-way ANOVA: 0 h, H_2_O_2_, *p* = 0.0001 and 1900 K, *p*  <  0.0001; 3 h, H_2_O_2_, *p*  <  0.0001, 1900 K, *p* = 0.0079, and 1900 K + H_2_O_2_, *p* = 0.005; 9 h, H_2_O_2_, *p* =  0.0012 and 1900 K, *p* = 0.0001; 24 h, H_2_O_2_ and 1900 K + H_2_O_2_, *p*  <  0.0001 and 1900 K, *p* = 0.0003); ^#^ significantly different from H_2_O_2_ (one-way ANOVA: 0 h, 1900 K, *p*  <  0.0001 and 1900 K + H_2_O_2_, *p* =  0.0032; 3 h, 1900 K and 1900 K + H_2_O_2_, *p*  <  0.0001; 9h, 1900 K, *p*  <  0.0001 and 1900 K + H_2_O_2_, *p* =  0.0009; 24 h, 1900 K and 1900 K + H_2_O_2_, *p*  <  0.0001); ^+^ significantly different from 1900 K (one-way ANOVA: 0 h, 1900 K + H_2_O_2_, *p*  <  0.0001; 3 h, 1900 K + H_2_O_2_, *p*  <  0.0001; 9 h, 1900 K + H_2_O_2_, *p* =  0.0002; 24 h, 1900 K + H_2_O_2_, *p*  <  0.0001). The morphology of cells (**g**) and cell apoptosis detection (**h**) in the light pretreatment paradigm; concentration of H_2_O_2_ = 1000 μM. (**i**) Quantitative analysis of apoptosis rate; *n* = 3. * Significantly different from dark group (one-way ANOVA, *p*  <  0.0001); ^#^ significantly different from H_2_O_2_ group (one-way ANOVA, *p*  <  0.0001); ^+^ significantly different from 1900 K group (one-way ANOVA, *p*  <  0.0001). The data are presented as the mean ± SD. Bars in the bright field, 50 μm.

**Figure 3 ijms-24-04794-f003:**
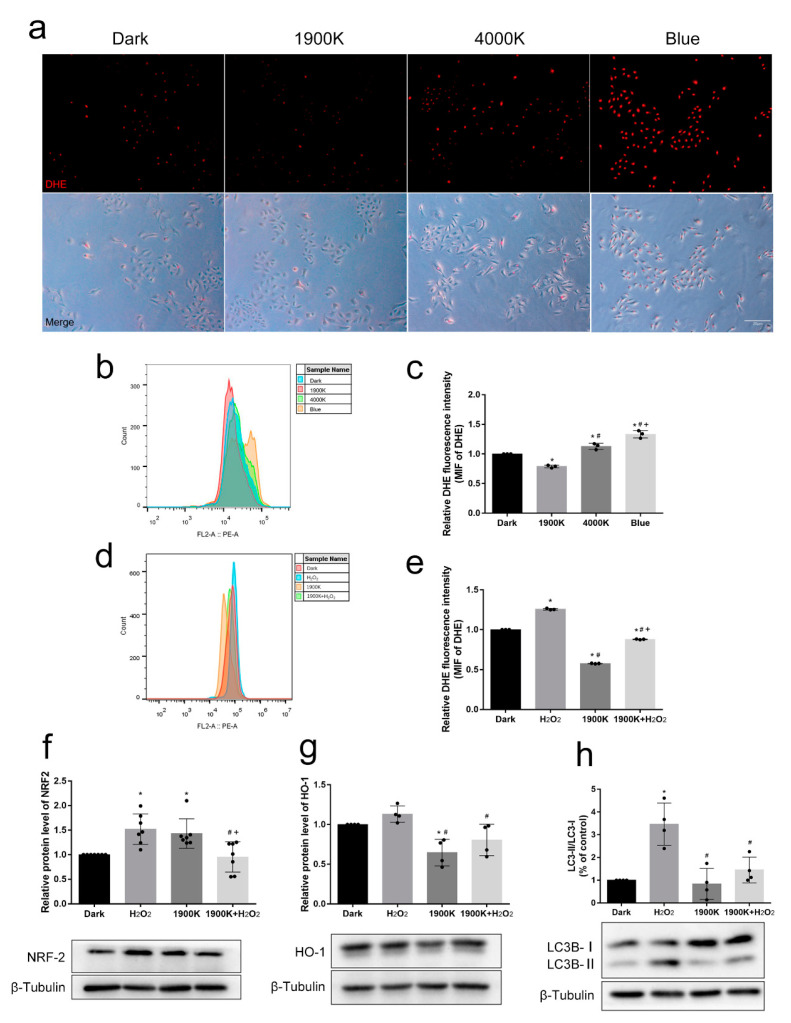
The 1900 K LEDs play an active role in the antioxidative stress of ARPE-19 cells. (**a**) The fluorescence of dihydroethidium (DHE) after irradiation with 1900 K, 4000 K, and blue-light LEDs for 48 h. The number and intensity of fluorescence represent the level of intracellular reactive oxygen species (ROS). The top panel shows the red fluorescence of DHE, and the lower panel consists of merged images with the bright field; *n* = 3. (**b**,**c**) Quantitative fluorescence analysis of dark, 1900 K, 4000 K, and blue-light LED groups; *n* = 3. * Significantly different from dark group (one-way ANOVA: 1900 K, *p* = 0.0013; 4000 K, *p* = 0.0237; blue, *p*  <  0.0001); ^#^ significantly different from 1900 K (one-way ANOVA, *p*  <  0.0001); ^+^ significantly different from 4000 K (one-way ANOVA, *p* =  0.0016). (**d**,**e**) Quantitative fluorescence analysis of dark, H_2_O_2_, 1900 K, and 1900 K + H_2_O_2_ groups of light pretreatments involving 1900 K LED pre-illumination for 48 h and then 400 μM H_2_O_2_ damage for 6 h; *n* = 3. * Significantly different from dark group (one-way ANOVA, *p*  <  0.0001); ^#^ significantly different from H_2_O_2_ group (one-way ANOVA, *p*  <  0.0001); ^+^ significantly different from 1900 K group (one-way ANOVA, *p* < 0.0001). (**f**–**h**) The levels of nuclear factor E2-related factor 2 (NRF2) (*n* = 7), heme oxygenase-1 (HO-1) (*n* = 4), and microtubule-associated protein 1 light chain 3 beta (LC3B) (*n* = 4) were detected by Western blot analysis in four groups of light pretreatments. β-tubulin was used as an endogenous control. * Significantly different from dark group (one-way ANOVA: NRF2, H_2_O_2_, *p* = 0.0062 and 1900 K, *p*  =  0.0271; HO-1, 1900 K, *p* = 0.0167; LC3B, H_2_O_2_, *p* =  0.0007); ^#^ significantly different from H_2_O_2_ group (one-way ANOVA: NRF2, 1900 K + H_2_O_2_, *p* =  0.0028; HO-1, 1900 K, *p* = 0.0017 and 1900 K + H_2_O_2_, *p*  =  0.0280; LC3B, 1900 K, *p*  =  0.0004 and 1900 K + H_2_O_2_, *p* =  0.0038); ^+^ significantly different from 1900 K (one-way ANOVA, NRF2, 1900 K + H_2_O_2_, *p*  =  0.0127). The data are presented as the mean ± SD. Bars in the merged images, 20 µm.

**Figure 4 ijms-24-04794-f004:**
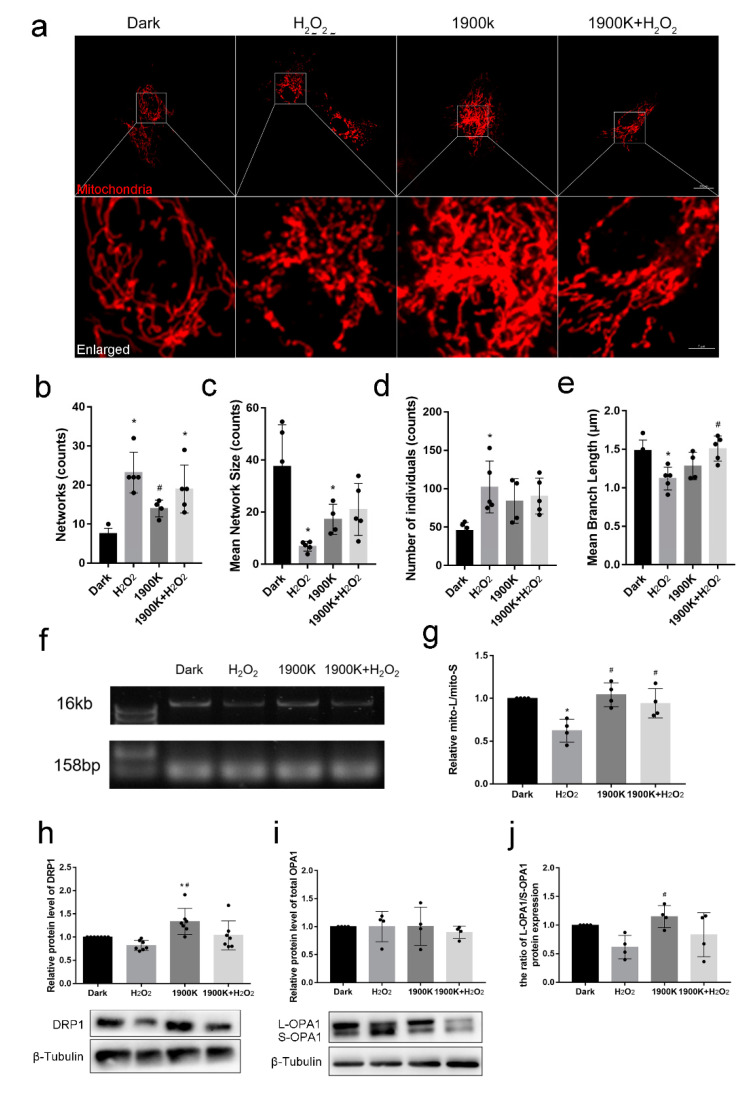
The 1900 K LEDs can reduce mitochondrial damage caused by H_2_O_2_ in ARPE-19 cells. (**a**) Fluorescence images show the alteration in mitochondrial morphology after irradiation with the 1900 K LEDs for 48 h, followed by 400 μM H_2_O_2_ damage for 3 h; *n* = 5. The top panel shows fluorescence images of mitochondrial morphology after the indicated treatments. The lower panel consists of partial enlarged view of each group. (**b**) The number of mitochondrial networks; *n* = 5, 5, 4, and 5. * Significantly different from dark group (one-way ANOVA: H_2_O_2_, *p* = 0.0002; 1900 K + H_2_O_2_, *p* = 0.0042); ^#^ significantly different from H_2_O_2_ group (one-way ANOVA: 1900 K, *p* = 0.0293). (**c**) The number of mitochondrial network size (branches); *n* = 5, 5, 4, and 5. * Significantly different from dark group (one-way ANOVA: H_2_O_2_, *p* = 0.0012; 1900 K, *p* = 0.0395). (**d**) The number of mitochondrial individuals; *n* = 5, 5, 4, and 5. * Significantly different from dark group (one-way ANOVA: H_2_O_2_, *p* = 0.0151). (**e**) The mean length of mitochondrial network branches; *n* = 5, 5, 4, and 5. * Significantly different from dark group (one-way ANOVA: H_2_O_2_, *p* = 0.0097); ^#^ significantly different from H_2_O_2_ group (one-way ANOVA: 1900 K + H_2_O_2_, *p*  =  0.0060). (**f**) The representative gels of mitochondrial DNA damage assay. Long mitochondrial segments and short mitochondrial segments had different band densities of about 16,568 bp and 158 bp, respectively. (**g**) The quantitative analysis of the ratio of mitochondrial DNA long chain to short chain; *n* = 4. * Significantly different from dark group (one-way ANOVA: 1900 K, *p* = 0.0066); ^#^ significantly different from H_2_O_2_ group (one-way ANOVA: 1900 K, *p* = 0.0030; 1900 K + H_2_O_2_, *p*  =  0.0197). (**h**) The protein level of dynamin-related protein 1 (DRP1); *n* = 7. * Significantly different from dark group (one-way ANOVA: 1900 K, *p* = 0.0361); ^#^ significantly different from H_2_O_2_ group (one-way ANOVA: 1900 K, *p* = 0.0009). (**i**) The relative protein level of total optic atrophy protein 1 (OPA1); *n* = 4. (**j**) The ratio of L-OPA1/S-OPA1; *n* = 4. ^#^ Significantly different from H_2_O_2_ group (one-way ANOVA: 1900 K, *p* = 0.0348). The data are presented as the mean ± SD. Bars in the original fluorescence images, 20 μm. Bars in the enlarged fluorescence images, 5 μm. Abbreviations: mito-L, mitochondrial DNA long chain; mito-S, mitochondrial DNA short chain.

**Figure 5 ijms-24-04794-f005:**
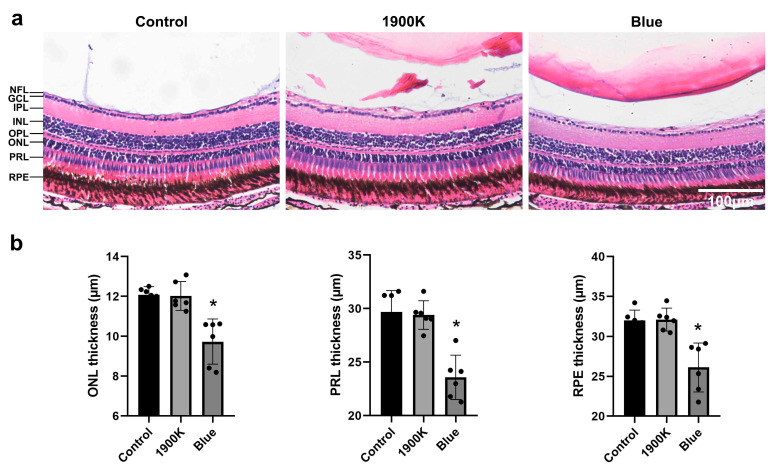
Irradiation with the 1900 K LEDs does not cause retinal damage in zebrafish. (**a**) Hematoxylin and eosin stained retinal tissues of normal zebrafish and experimental zebrafish exposed under 3000 lux 1900 K LEDs and blue-light LEDs for 7 days. (**b**) Quantification of outer nuclear layer (ONL), photoreceptor layer (PRL), and retinal pigment epithelium (RPE) thickness of normal zebrafish and experimental zebrafish exposed under 3000 lux 1900 K LEDs and blue-light LEDs for 7 days. * ONL thickness significantly different from control group (one-way ANOVA: blue, *p* = 0.0005). * PRL thickness significantly different from control group (one-way ANOVA: blue, *p* = 0.0001). * RPE thickness significantly different from control group (one-way ANOVA: blue, *p* = 0.0006). The quantitative results of ONL, PRL, and RPE thickness per vertical column are presented as the mean ± SD (*n* = 6 eyes in each group). Scale bar = 100 μm. Abbreviations: NFL, nerve fiber layer; GCL, ganglion cell layerIPL, inner plexiform layer; INL, inner nuclear layer; OPL, outer plexiform layer.

**Figure 6 ijms-24-04794-f006:**
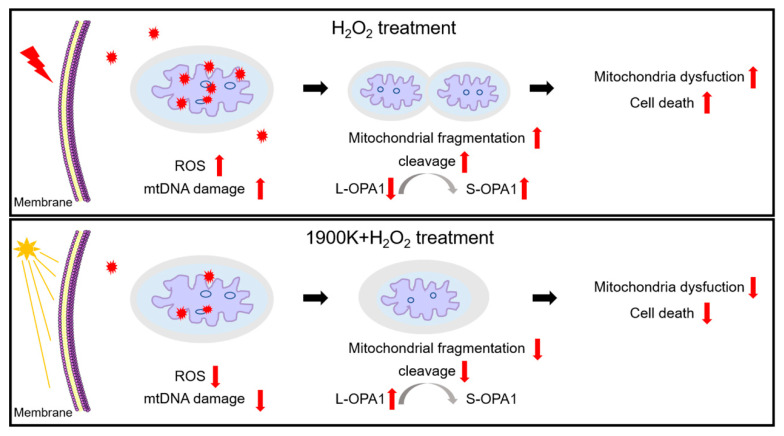
The role of 1900 K LEDs in H_2_O_2_-induced damage in RPE. H_2_O_2_ increased intracellular ROS levels, mitochondrial fragmentation, and mitochondrial DNA damage, which led to mitochondrial dysfunction and, ultimately, cell death. Pretreatment with the 1900 K LEDs could reduce intracellular ROS levels, mitochondrial fragmentation, and mitochondrial DNA damage, protect mitochondrial function, and ultimately prevent cell death.

**Figure 7 ijms-24-04794-f007:**
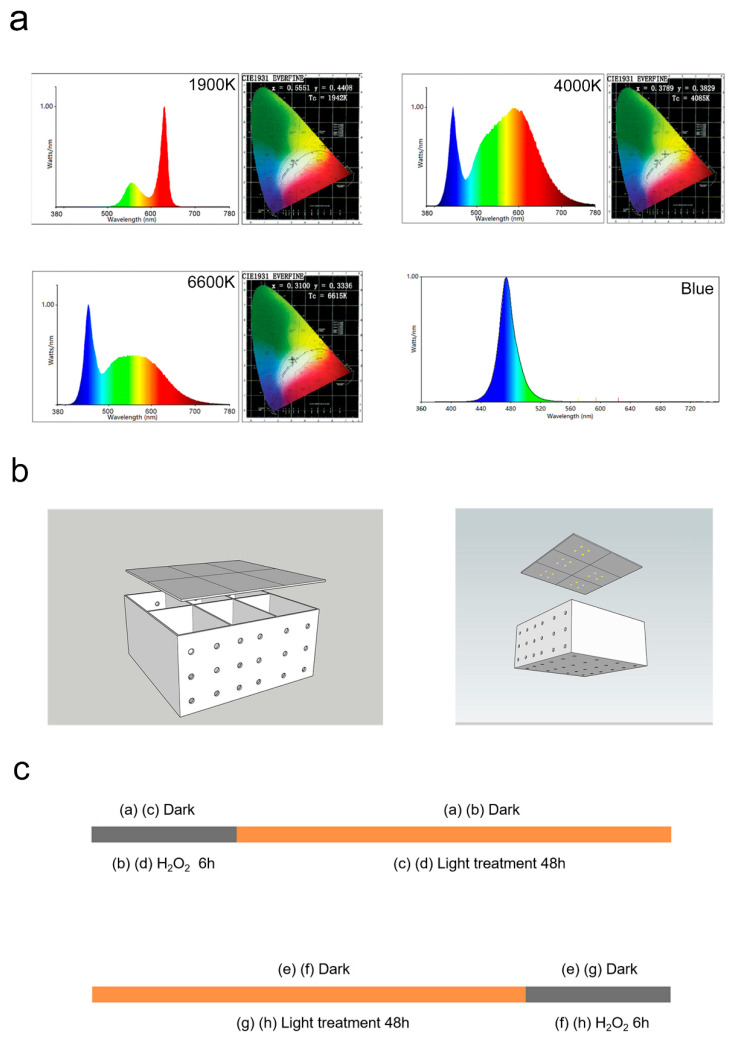
Illuminating device and methodological timeline. (**a**) The 1900 K LEDs have two peaks at 560 and 630 nm and no blue content. The 4000 K and 6600 K LEDs have a peak at 450 nm, followed by a second peak representing yellow light. The amplitude of 6600 K LEDs is lower than that of 4000 K LEDs. Blue-light LEDs only have a single wave peak at 470 nm. (**b**) The illuminating device consisted of a lamp cover and a box divided into six independent areas. We put lamps on the inner surface of the lamp cover. Air holes were designed on noncontact surfaces of six different areas to avoid light interference. This study mainly used two modeling methods (**c**): the light post-treatment paradigm and the light pretreatment paradigm. There were four groups in each modeling method (dark group, H_2_O_2_ group, 1900 K LED group, and H_2_O_2_ + 1900 K group or 1900 K + H_2_O_2_ group). The light post-treatment paradigm involved damage from 400 μM H_2_O_2_ for 6 h, followed by light treatment with the 1900 K LED for 48 h. The other paradigm involved light treatment with the 1900 K LED for 48 h followed by damage from 400 μM H_2_O_2_ for 6 h. Abbreviations: LED, light-emitting diode; H_2_O_2_, hydrogen peroxide.

**Table 1 ijms-24-04794-t001:** Primary antibodies used in the study.

Antibody	Source	Catolog.No	Type of Ab	Dilution	MW
HO-1	Abcam	Ab13248	Mouse mAb	1:1000	34.6
NRF2	Abmart	T55136F	Rabbit mAb	1:1000	110
LC3B	CST	2775s	Rabbit mAb	1:1000	14/16
DRP1	CST	#8570	Rabbit mAb	1:1000	78–82
OPA1	BD	612606	Mouse mAb	1:1000	80–100
β-tubulin	TRANS	HC101-01	Mouse mAb	1:2500	55

## Data Availability

The original contributions presented in the study are included in the article/Appendix A; further inquiries can be directed to the corresponding author.

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
