# Peer review of "Antioxidative and Mitochondrial Protection in Retinal Pigment Epithelium: New Light Source in Action"

_ijms, 2023, doi:10.3390/ijms24054794_

Round 1

Reviewer 1 Report

Dear Authors,

You have done a great job.  It is an interesting manuscript by design that should be published.  Each section of the manuscript is well documented, with results that are clearly presented and comparisons made with the existing literature. The English is acceptable and the references are relatively recent. However, please attention to some font inconsistencies. This study includes Ethical Approval and the iconography is in line with the content.

Kind regards and all the best,

The Reviewer

Reviewer 2 Report

The article by Ming Jin et al is devoted to the study of the role of low-color-temperature light-emitting diodes (LEDs) (1900 K LEDs) on retinal pigmented epithelium. In their work, the authors show that the 1900 KLEDs pre-treatment could protect RPE from death after H2O2 damage. In addition, the 1900 K LEDs irradiation in zebrafish did not cause retinal damage within 7 days.

Comments:

1.      I would like to understand what the authors attribute to the increase in the survival of retinal pigment epithelium cells after treatment with 1900 K LED (Fig. 1). The survival rate cannot be more than 100%. If the authors do not deny cell proliferation, then it is necessary to evaluate it and determine the number of cells. Moreover, if proliferation is observed, did the authors take into account the number of cells in subsequent experiments?

2.      It is not very clear how the authors normalized the amount of protein in WB. It can be noted that the level of tubulin in many figures (file with WB) is different, which does not allow to reliably characterize the changes observed by the authors. Also, why did the authors normalize control protein amounts to 100% without representing variation? Why are proteins (eg OPA1 and Tubulin) visualized using different methods (group 2 and 3 OPA1)?

3.      The authors report that 1900 K LED prevents mitochondrial fragmentation in the presence of H2O2 (fig. 6). However, according to the data in Fig. 4, 1900K LED increases the level of DRP1, which is responsible for cell division. What do the authors attribute this fact to? Authors should assess the number and morphology of mitochondria. They have the ability to do so.

Reviewer 3 Report

In this paper, the authors evaluated the protective effects of 1900 K LEDs on retinal pigment epithelium (RPE) cells. The investigated the effect of H2O2 on the cells and how 1900 K LEDs could be protective by analysing cell activity, apoptosis, antioxidative stress markers, mitochondrial damage, and retinal damage in zebrafish.

The study is well designed, written and discussed. I have some few comments:

1) In paragraph 2.1.: the authors chose the irradiance of 10 W/m2 stating that the cell activity was the most significant in Figure 1b. However, there is no statistical analysis mentioned between the 3 different irrandiances. The cell activity at 10 w/m2 is indeed higher but also with higher variation. I suggest to perform a statistical analysis between the 3 irrandiances groups. 

2) Figure 2 c-f: The graphs represent the cell activity after removing H2O2 at different time points. I suggest to add the time points on the top of each graph to avoid confusion.

3) Figure 6 summarizes the role of 1900 K LEDs in the H2O2 model. This figure is not mentioned in the main text. 

4) Line 484-485-487: Figure 6 should be corrected to Figure 7

5) Line 60: PBM abbreviation is not defined

Round 2

Reviewer 2 Report

The manuscript is appropriately revised. But I think that the survival rate cannot be more than 100%. A cell activity is a more correct name for the Y axis.

Author Response

we have taken your advice and changed the name of the Y axis to cell
activity in the cell activity results. The relevant modifications relate to Figure 1 and
Figure 2a-2f.